# Electroneurographic Evaluation of Neural Impulse Transmission in Patients after Ischemic Stroke Following Functional Electrical Stimulation of Antagonistic Muscles at Wrist and Ankle in Two-Month Follow-Up

**DOI:** 10.3390/ijerph19020713

**Published:** 2022-01-09

**Authors:** Katarzyna Kaczmarek, Juliusz Huber, Katarzyna Leszczyńska, Przemysław Daroszewski

**Affiliations:** 1Neurology Ward, Pomeranian District Hospital, Chałubińskiego No. 7, 75-581 Koszalin, Poland; kasia@przystan.pl; 2Department of Pathophysiology of Locomotor Organs, University of Medical Sciences, 28 Czerwca 1956 No. 135/137, 60-545 Poznań, Poland; kat.leszczynska@gmail.com; 3Department of Organization and Management in Health Care, University of Medical Sciences, 28 Czerwca 1956 No. 135/137, 60-545 Poznań, Poland; dyrektor@orsk.ump.edu.pl

**Keywords:** ischemic stroke, rehabilitation, functional electrical stimulation, proprioceptive neuromuscular facilitation, electroneurographic studies

## Abstract

The available data from electroneurography (ENG) studies on the transmission of neural impulses in the motor fibers of upper and lower extremity nerves following neuromuscular functional electrical stimulation (NMFES) combined with kinesiotherapy in post-stroke patients during sixty-day observation do not provide convincing results. This study aims to compare the effectiveness of an NMFES of antagonistic muscle groups at the wrist and ankle and kinesiotherapy based mainly on proprioceptive neuromuscular facilitation (PNF). An ENG was performed once in a group of 60 healthy volunteers and three times in 120 patients after stroke (T0, up to 7 days after the incident; T1, after 21 days of treatment; and T2, after 60 days of treatment); 60 subjects received personalized NMFES and PNF treatment (NMFES+K), while the other 60 received only PNF (K). An ENG studied peripheral (M-wave recordings), C8 and L5 ventral root (F-wave recordings) neural impulse transmission in the peroneal and the ulnar nerves on the hemiparetic side. Both groups statistically differed in their amplitudes of M-wave recording parameters after peroneal nerve stimulation performed at T0 and T2 compared with the control group. After 60 days of treatment, only the patients from the NMFES+K group showed significant improvement in M-wave recordings. The application of the proposed NMFES electrostimulation algorithm combined with PNF improved the peripheral neural transmission in peroneal but not ulnar motor nerve fibers in patients after ischemic stroke. Combined kinesiotherapy and safe, personalized, controlled electrotherapy after stroke give better results than kinesiotherapy alone.

## 1. Introduction

Successful rehabilitation of patients after ischemic stroke consists of pharmacological treatment [1] and kinesiotherapeutic procedures based on the proprioceptive neuromuscular facilitation (PNF) method performed by physiotherapists according to the worldwide accepted algorithm [2,3,4]. Physical therapy applied to post-stroke patients consists mainly of therapy based on warming and electrotherapy of the paretic muscles. The aim is to improve the activity of the motor units undergoing pathological neurogenic change. Current trends in electrotherapy focus on the functional electrical stimulation of nerves (FES), and neuromuscular functional electrical stimulation (NMFES). Various combinations of stimuli algorithms have proven to be moderately effective [5]. NMFES can be applied to the muscles acting antagonistically at the wrist and ankle joints in post-stroke patients, as described by Lisiński et al. [6]. Moreover, Kraft et al. [7] similarly proved the effectiveness of the PNF and FES combination in patients after stroke; they presented the improvement of the function of the paretic muscles on the symptomatic side in both upper and lower extremities.

Up to now, a study on the effectiveness of NMFES combined with PNF therapy has been performed in a small group of patients after ischemic stroke (*N* = 24) in a short-term observation (after 20 days) [6]. The study with observations up to two months undertaken here may confirm the sustained effects of such a treatment, thereby fulfilling the crucial condition of rehabilitation continuity.

Changes in motor fiber transmission of neural impulses in post-stroke patients, as well as their etiology, have been described in a different way in previous studies. Abnormalities may be explained by the disturbances of the transmission of the efferent impulses from the supraspinal centers to the motor cells at the spinal levels. Pathologies in the peripheral motor transmission may appear despite no structural cause of damage at the spinal neuromeres. The lack of motoneuron excitation in post-stroke patients may influence degenerative axonal changes, which later propagate peripherally and, finally, evoke motor unit atrophy. Previous electroneurographic examinations have shown moderate improvement in neural transmission in the motor fibers of the upper and the lower extremity nerves following applied NMFES to muscles acting antagonistically at the wrist and the ankle in patients after an ischemic stroke [6]. However, the general consensus of other research is that the motor fibers of both the upper and the lower extremities, predominantly on the paretic side, showed more axonal than demyelinating abnormalities [8,9]. It can be hypothesized that objective, noninvasive, clinical neurophysiological studies seem to help improve the quality of antagonistic muscle electrostimulation procedures in patients after stroke. Electroneurography (ENG) precisely evaluates the transmission of the neural impulses in nerves, while surface electromyography (sEMG) assesses quantitatively and qualitatively the activity of muscle motor units; both have been rarely used for functional evaluation in the studies on post-stroke patients.

The study aims to evaluate the effectiveness of an NMFES algorithm of antagonistic muscle groups acting at the wrist and the ankle combined with kinesiotherapy treatment based mainly on proprioceptive neuromuscular facilitation (PNF) and compare it with treatment based mainly on PNF performed in a rehabilitation center for ischemic stroke patients in a two-month follow-up. The measurement outcome was the transmission of motor neural impulses in fibers of the ulnar and peroneal nerves on the paretic side in comparison to the results recorded in healthy volunteers and among the two studied groups.

## 2. Materials and Methods

### 2.1. Subjects and Study Design

We recruited 145 ischemic stroke subjects and 60 healthy volunteers for the study (Figure 1). The main inclusion criteria for the patients were age (45–70 years), and a clinically confirmed ischemic stroke on a CT or MRI scan performed immediately at the acute phase (T0). Then, patients were monitored at the subacute phase (T1) and later at the subacute phase (T2), a period of early post-stroke rehabilitation, for not less than 60 days (Table 1).

The main contraindications for the participation in the project were epilepsy or previous consequences of ischemic stroke, severe disorders of the cardiovascular system, pregnancy, electronic implants such as pacemakers and cochlear implants, inflammatory diseases, proximal and distal neuropathies episodes in treatment (including COVID-19 related), or myelopathies before the hospitalization. The patients had to give written consent for participation in the project for not less than six months; however, seven did not meet the inclusion criteria, five declined to participate or died. All patients understood the potential of no benefit and were informed of the risk of all procedures. Of the 133 patients, 67 who agreed to participate in the project with electrotherapy procedures and did not present contraindications for electrostimulation were recruited to the NMFES+K group treated with kinesiotherapy and muscle electrotherapy under the supervision of a team of the physical and rehabilitation medicine physician and the physiotherapist. The other sixty-six patients received only kinesiotherapeutic treatment (K group) with a similar rehabilitation program as those from the NMFES+K. These patients either did not meet the inclusion criteria or they strongly refused application of the electrostimulation procedures. At about the third day after the patients have been admitted in the neurological ward, they were generally stable enough to start with the physiotherapeutic treatment. Until the end of the T2 observation period, three patients of the K group and three of the NMFES+K group could not continue due to COVID-19 hospitalization or death. We randomly decreased the number of patients (three in the K group and four in the NMFES+K group) for the final analysis, which included 60 patients in both groups (Figure 1). Both NMFES+K and K groups showed similar symptoms of ischemia in subcortical (55%) (Figure 2A) or frontoparietal (45%) (Figure 2B) areas on CT or MRI scans. The cross-sectional (coronal) area of ischemia averaged 276 mm^2^ ± 65 mm^2^ in the K group and 297 mm^2^ ± 82 mm^2^ in the NMFES+K group.

Table 1 presents the anthropometric characteristics of the healthy subjects and the patients, who did not differ significantly in age, height, or weight. The same set of neurophysiological studies, the results of which verified the effectiveness of therapy in two groups of patients, was applied once to the group of 60 healthy volunteers (a control group) to obtain reference values (Table 3). Electroneurography recording (ENG) of motor ulnar and peroneal nerve fibers in the more paretic side (evaluated in sEMG, and Lovett’s scale tests) was used to analyze how kinesiotherapy combined with electrostimulation or kinesiotherapy as a single method of treatment influenced the health status of patients belonging to the two groups. The study was conducted according to the guidelines of the Declaration of Helsinki and approved by the Bioethics Committee of the University of Medical Sciences (Decision No. Resolution 1279/18).

### 2.2. Neurophysiological Evaluation

The neurophysiological testing was performed with the Keypoint System (Medtronic A/S, Skovlunde, Denmark). Electroneurographic recordings were performed with the same rules before treatment (T0) and at two stages of observations (T1, after 21 days and T2, after two months of treatment) to evaluate the neural transmission in the motor fibers of ulnar and peroneal nerves in both groups (Figure 3a,b, respectively). The strength of stimuli applied to the nerves during ENG studies to evoke the maximal amplitude of motor-evoked potential on the orthodromic way (M-wave, CMAP potential) was used for the personally adjusted electrostimulation algorithm (Table 2). Electrical rectangular pulses with 0.2 ms duration, at 1 Hz, and the intensity from 0 to 80 mA delivered from the bipolar stimulating electrodes were applied bilaterally to the nerves in three positions according to their anatomical passages over the skin. The pairs of surface electrodes recorded evoked potentials from abductor digiti minimi and extensor digitorum longus muscles; the same surface electrodes were also used for sEMG recordings (see below). We analyzed the parameters of amplitude (in µV) and latency (in ms) in M–wave ENG recordings. The recordings were performed at the amplification of 100 to 5000 (µV) and a time base of 8 ms and compared to the normative values recorded in the healthy volunteers with the patients of both groups (Table 1 and Figure 5). M-wave amplitude and corresponding conduction velocities of nerve impulses were calculated to assess peripheral neural transmission in nerve fibers. F-wave frequencies (during evoking 20 positive successive recordings of M waves) were analyzed to ascertain neural motor transmission in C6–C7 and L5 ventral roots [10,11]. The frequency of antidromically evoked 14 F-waves is accepted as the proper neural transmission from the level of the motoneuron to the effector.

### 2.3. Treatment with NMFES

Antagonistic muscle groups acting at the wrist and the ankle of patients from the NMFES+K group were stimulated using a personal, mobile, four-channel device (NeuroTrac^®^ Sports XL, Verity Medical Ltd., Hampshire, UK). The general principles of NMFES were based on the description of a method proposed in the work of Lisinski et al. [6] with modifications developed in Huber et al. [12]. During both sEMG recordings and NMFES, we used the same locations of electrodes covering the location of the motor points of the extensor carpi muscle group versus the flexor carpi muscle group (at the wrist) and the tibialis anterior muscle versus calf muscle group (medial, lateral and soleus muscles at the ankle) (Figure 4). Two pairs of self-adhesive surface electrodes (Axelgaard Ultrastim Wire Neurostimulation Electrodes with MultiStick Gel, 5 cm × 5 cm, Axelgaard Manufacturing Co. Ltd., Lystrup, Denmark) were placed on the skin over the anatomical position of the muscle. The cathode was placed on the distal tendon of the muscle, while the anode was placed on the muscle belly. Electrostimulation was applied in an alternative mode, which means that the stimulation device released via two pairs of bipolar surface electrodes the trains of electrical stimuli exciting first flexors and then extensor muscle groups at the wrist and the ankle (Figure 4Ad,Bd).

The sEMG recordings from the abovementioned muscles and ENG results obtained at T0 were both used to create the individually adjusted algorithm of electrostimulation applied to the patients from the NMFES+K group (Table 2). The variables of the stimulation algorithm were set up as follows: the frequency of stimuli (electrical bipolar rectangular pulses, with subsequently upper-lower inflexions and negative-positive according to the neurophysiological terminology) in one train delivered from the electrodes depended on the sEMG frequency parameter recorded during an attempt at a maximal muscle contraction (35–70 Hz, 48.6 Hz on average); the interval between the bursts of pulses was from 2 to 5 s (4.3 s on average); the single stimulus duration was calculated from the repetitive measurements of the successive duration of single muscle motor action potentials in the sEMG recordings (14.1 ms on average); and the stimulus intensity set up for the muscles of the upper and lower extremities was 26.8 mA on average and calculated from the stimulus strength applied to the ulnar and peroneal nerves to evoke the maximal amplitude of an M-wave response. All the parameters were set up and supervised by a team of the physical and rehabilitation medicine physician and the physiotherapist. The stimulating device was blocked to prevent unplanned changes applied by the patients up to T1 when the algorithm could be verified and modified by the same team. The stimulus strength was the only parameter that could be changed by the participants. They were instructed to increase the stimulus strength during the single stimulation session to observe the visible contraction of the stimulated muscles and without intrusive pain. The duration of one session depended on the severity of neurogenic changes ascertained in sEMG recordings from 15 to 20 min (19.1 min on average). NMFES sessions were held with the frequency of five times a week for a period of not less than 2 months. The data in Table 1 indicate that patients were treated from 50 to 72 days (62 ± 6 days on average). The stimulating device allowed the storage of the settings in the memory. They could be read out at T1 and T2 to verify the therapy course. The calculated time of expected stimulation was almost equal to the detected time of stimulation. All the details regarding the stimulation parameters applied to the NMFES+K group have been presented in Table 2.

Another form of physical therapy consisted mainly of “warm-up therapy”; no electrotherapy treatment other than what was described in the study was performed.

### 2.4. Kinesiotherapy

The rehabilitation treatment was performed in an outpatient department and applied with the same regime in both the NMFES+K and K groups of patients. Kinesiotherapy provided by the physiotherapist was mainly based on the PNF method. According to Adler et al. [4], the therapists applied the PNF stretching algorithm that consisted of flexion, abduction, and external rotation as well as extension, abduction, and internal rotation for the paralyzed upper and lower extremities, respectively. The physiotherapists also applied individualized passive, supportive, and active muscle exercises on the paretic side in addition to exercises that decreased the spasticity symptom (post-isometric relaxation (PIR) procedures) and stretching exercises to stimulate proprioceptors according to the standard post-stroke rehabilitation treatment. One daily kinesiotherapeutic treatment session lasted about 3 h. The patients from the NMFES+K group were treated with kinesiotherapy 62 ± 6 days on average, while patients from the K group were treated 63 ± 6 days on average. Both groups received daily treatment except on Saturdays and Sundays. During the 10-day stay at the neurology ward, patients were upright. A physiotherapist instructed, supported, and motivated the patient to sit down, change body positions to prevent bed sores, and perform other global movements necessary for daily living (e.g., change of position in bed or transferring from a wheelchair to a chair). Another element of rehabilitation was developing locomotion, including walking with a handrail, and walking with the use of orthopedic equipment, such as a walker or a walking stick.

### 2.5. Data Analysis

Data were analyzed with Statistica, version 13.1 (StatSoft, Kraków, Poland). For measurable variables, descriptive statistics included mean and median values, standard deviations (SD), and minimum (min) and maximum (max) values. In some cases, Shapiro–Wilk tests and Levene’s tests were used to conduct the normality distribution and homogeneity of variances. We used the Student’s *t*-test to compare the median values of parameters from neurophysiological studies in 120 patients with ischemic stroke; in some cases, ANOVA was used. A comparison of values at *p* ≤ 0.05 was determined to be statistically significant. A preliminary analysis performed before the study was completed revealed the required sample size using the primary outcome variables from ENG recordings before and after treatment with a certainty of 80% and a significance level of 0.05 (two-tailed). The data from the first 15 subjects were used to calculate the mean and standard deviation (SD). The sample-size software estimated the minimum of subjects to be 45.

## 3. Results

We compared the total expected time of the stimulation with the detected realtime read out from the stimulation device (Table 2). In general, the post-stroke patients followed the electrostimulation regime. The average expected time was 19.2 h, while the average detected time was 18.4 h. None of the patients reported side effects from NMFES, including pain.

Examples of neurophysiological recordings performed at certain stages of observation before and after the applied treatment in both patient groups as well as in the healthy volunteers are shown in Figure 5; the ENG results are presented in Table 3.

The ENG recordings of the M-waves following the electrical stimulation of the ulnar nerves did not provide evidence of changes in motor fibers of the axonal (the decrease in the amplitude parameter) or the demyelinating types (the increase in the latency parameter) before and after treatment. They also did not show changes in the frequencies of the recorded antidromically evoked F-waves, which proves no functional changes in the neural transmission of C6–C7 ventral roots. The expected frequency of F-waves should be more than 14 potentials following 20 applied stimuli evoking constant M-waves in normal conditions. In general, post-stroke patients of both groups did not differ from the healthy subjects in the peripheral neural transmission of motor fibers in the ulnar nerves before and after treatment.

On the other hand, the comparison of measured amplitudes (but not the latencies) in the M-waves following electrical stimulation of the peroneal nerves differed between healthy subjects and patients before and after treatment. However, the improvement of the amplitude but not latency parameter has been observed between the T0 and T2 period of observation in the NMFES+K group (*p* = 0.04; Table 3), indicating the motor transmission recovery within the axons of the stimulated nerve branches. ANOVA analysis showed that the ENG M-wave amplitude change appeared even at T1 with only a slightly better improvement at T2. The graphic presentation of this improvement is also presented in Figure 6. Such changes were not observed in the K group at T2. Moreover, K group post-stroke patients showed a statistically significant decrease in the F-wave frequencies (*p* = 0.04; Table 3), indicating worsening of the neural transmission of the L5 ventral roots motor fibers.

## 4. Discussion

To date, the available literature does not provide results of similar studies describing the effects of personalized 60-day NMFES applied to the antagonistic muscles at the wrist and the ankle conjoined with PNF therapy in post-stroke patients. So far, only Lisiński et al. [6] has proven the effectiveness of a rehabilitation therapy evaluated using neurophysiology methods, such as electroneurography, on the transmission of nerve impulses in upper and lower extremities but only in a 20-day observation. They found post-stroke abnormalities in the transmission of neural impulses both in the ulnar and peroneal motor nerve fibers; the current study did not reveal changes in the upper extremity nerves on the paretic side in T0. These differences between results of the previous and the current study might be explained by the dissimilar functional statuses of post-stroke patients before the treatments were undertaken and the different structural symptoms of ischemic changes at the supraspinal level. In fact, in this study, slight but statistically insignificant changes in the parameters of ENG studies following the stimulation of the ulnar nerves were found between the patients and healthy subjects (Table 3). Other previous studies using a similar ENG methodology, described only the abnormalities in the peripheral neural transmission of the fibers in more upper than the lower extremity nerves, but electroneurography was used in post-stroke patients for diagnostic purposes, without assessment of the applied treatment [8,9].

The reason for the abnormalities in the transmission of impulses in the peripheral motor fibers of the nerves in the lower extremities may be explained by the disturbances of the transmission of the efferent impulses from the supraspinal centers to the motor cells at the spinal levels. We found pathologies in peripheral motor transmission despite the lack of any structural cause of damage at the spinal neuromeres. Moreover, the influence of age, degenerative spinal disease, and lumbosacral root and disc conflict could be considered as the causes of this condition. The last conclusion is based on a significantly reduced frequency of F-wave responses evoked from the peroneal nerves in electroneurographic studies. The reduced frequency with 15–16 recorded F-wave responses in patients compared to 19 recorded in healthy subjects at *p* = 0.04 may confirm changes in the neuronal transmission of L5 ventral root motor fibers.

It is worth mentioning, considering the positive results of electrotherapy applied in the NMFES+K group patients, that a team of the physical and rehabilitation medicine physician and the physiotherapist was able to set the parameters, save, and secure the settings to prevent unplanned changes made by the participants. This had a significant impact on the patient, who, when under control, followed the stimulation regime more carefully.

The other reason for the positive effects in NMFES+K group is that the algorithm of the stimulations was individually adjusted to the functional post-stroke patients’ status of peripheral neural transmission and the muscle motor units’ activity acting antagonistically at the wrist and the ankle. We mainly used the frequency of the applied pulsons at 48.6 Hz. Similarly, Sentandreu-Mañó et al. [13] and Sabut et al. [14] used 35 and 50 Hz in a large population of patients during eight-week and 12-week programs, in which NMFES and stretching exercises were superior to conventional rehabilitation alone. However, instead of ENG studies, they performed clinical tests, which revealed reduced spasticity and improvement in wrist extensor activity in post-stroke patients. Similarly, the same effect was described by Eraifej et al. [15] on the development of daily living activities; moreover, they have stated the need for future research towards optimal parameters of standardized electrotherapy treatment.

The results of PNF stretching on the recovery of motor function in post-stroke patients are reported as twofold. Anas et al. [16] showed heterogeneous evidence of PNF interventions in patients after stroke, while Guiu-Tula et al. [17] stressed incorporating PNF stretching to functional training in stroke survival. In our study, the patients treated only with PNF (K group) showed a statistically significant worsening of ENG results following the stimulation of the peroneal nerves (Figure 6).

The convincing conclusion of Sahin et al. [18] and Guo et al. [19] about the effectiveness of NMFES applied together with PNF stretching on the wrist extensor muscle activity in post-stroke patients, similarly to our study, may be explained by the facilitation of two sources of afferent stimulations on the reflex motor recovery at the spinal or even the supraspinal levels, e.g., provided from NMFES electrotherapy and PNF stretching exercises. This explanation holds true, taking into account that the activity of the antagonistic muscle acting at the wrist and the ankle is regulated by the spinal Ia inhibitory interneuron reflex on a biofeedback way. The latter is, however, under the strong excitatory and inhibitory control of the impulses taking the origin from the corticospinal tract fibers and the propriospinal pathways [20], which are influenced by PNF procedures according to the suggestions of Sharman et al. [21].

One of the study limitations could be the duration of observation, which lasted only two months. Our intention was to perform the reliable treatments and observations, which were directly supervised by the same team of the physical and rehabilitation medicine physician and the physiotherapist, to avoid patients’ absences or receiving the heterogeneous or incomplete sets of rehabilitation procedures to both groups of subjects in the rehabilitation center belonging to the neurology ward in the hospital. However, the maximal duration of such medical care financed from the insurance sources lasts no longer than two months. Later patients are treated in the different outpatient clinics, which could somewhat influence the treatment regime, and, as we suspected, possibly the final results. Despite this, the undertaken study with observations up to two months confirmed the sustained effects of NMFES+K treatment. We also considered the pain sensed by the treated post-stroke patients during personally increasing the strength of NMFES stimuli, to reach the visible reaction of the effector, as the variable factor possibly influencing the study results, although it was the only patient-depended parameter. We are aware of some discrepancies in between the mean age of the control group and of the two groups of enrolled post-stroke patients (NMFES+K and K groups, Table 1), although, the latter were out of significance. However, the differences in results found in T0 and T2 in post-stroke patients of both groups and the healthy volunteers were still so significant (Table 3), to differentiate them from the status of healthy people, which proves that the mean age was not the most important factor in a total analysis performed in the current study.

The clinical importance of this study arises from the possibility of ascertaining the effectiveness of treatment, especially the method of functional muscle electrostimulation, on the principle of biofeedback in patients after a cerebral ischemic stroke. From the cognitive point of view, the study may provide knowledge about the abnormalities in peripheral motor neural transmission. Future sEMG studies on the NMFES effectiveness may reveal, indirectly, the phenomena of functional reorganization in the neuronal centers of the spinal cord and neuroplasticity at the supraspinal level.

## 5. Conclusions

After 60 days of treatment, only those post-stroke patients treated with neuromuscular functional electrical stimulation and kinesiotherapy improved significantly in the ENG results of M-wave recordings following the stimulation of the peroneal nerves. Kinesiotherapy combined with safe, personalized, controlled electrotherapy in patients after a stroke gives better results than kinesiotherapy alone.

## Figures and Tables

**Figure 1 ijerph-19-00713-f001:**
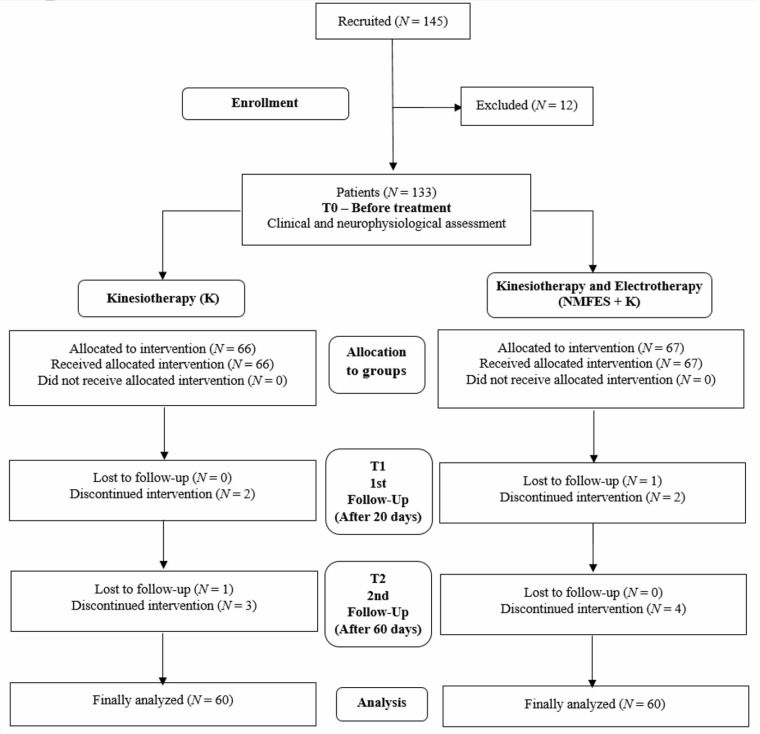
Flow chart of the study.

**Figure 2 ijerph-19-00713-f002:**
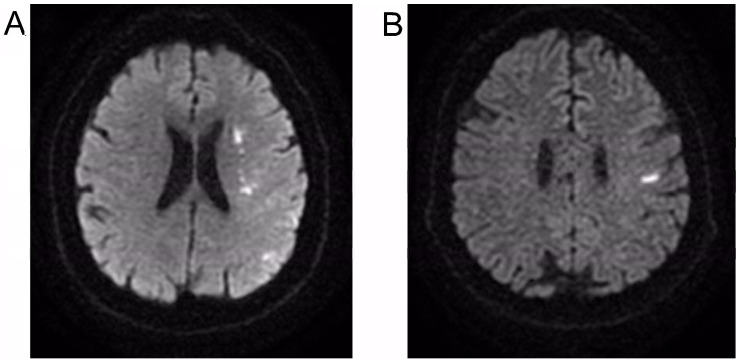
Examples of MRI pictures in DWI sequences presenting acute ischemic areas in patients from the NMFES+K group (**A**) and K group (**B**).

**Figure 3 ijerph-19-00713-f003:**
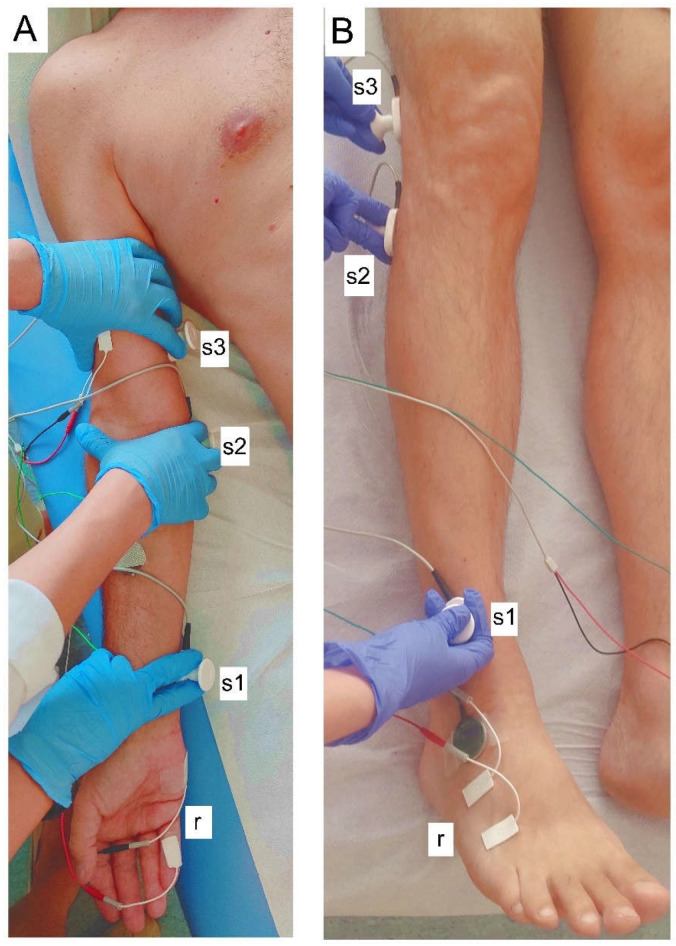
Principles of neurophysiological ENG studies in healthy volunteers and post-stroke patients. Figures show the sites of electrical stimulations via bipolar electrodes (s1–s3) and recordings (r) during ENG studies of neural transmission in the ulnar nerve (**A**) (s1, at the wrist; s2, at the ulnar sulcus; s3, above the elbow; and r, from abductor digiti minimi muscle) and peroneal (**B**) (s1, at the ankle; s2, below popliteal fossa; s3, above the popliteal fossa; and r, from extensor digitorum brevis muscle) nerves.

**Figure 4 ijerph-19-00713-f004:**
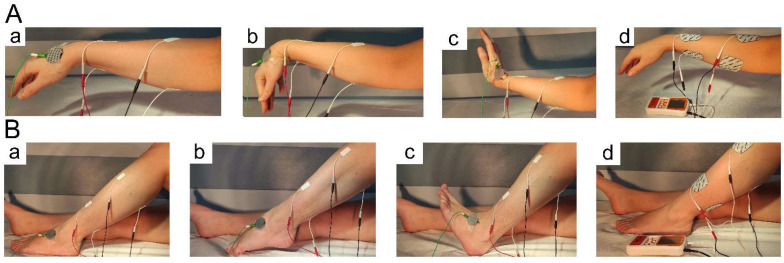
Placement of sEMG recording electrodes over the skin of the muscles acting antagonistically on the wrist (**Aa**–**c**) and the ankle (**Ba**–**c**). The position of NMFES stimulating electrodes (**Ad**, **Bd**) was the same; the active electrode was placed on the muscle belly, while the reference electrode was on the distal muscle tendon, covering the location of the motor points: (**a**), neutral position of the muscles at rest; (**b**), recordings during contractions of flexors; and (**c**), recordings during contraction of extensors.

**Figure 5 ijerph-19-00713-f005:**
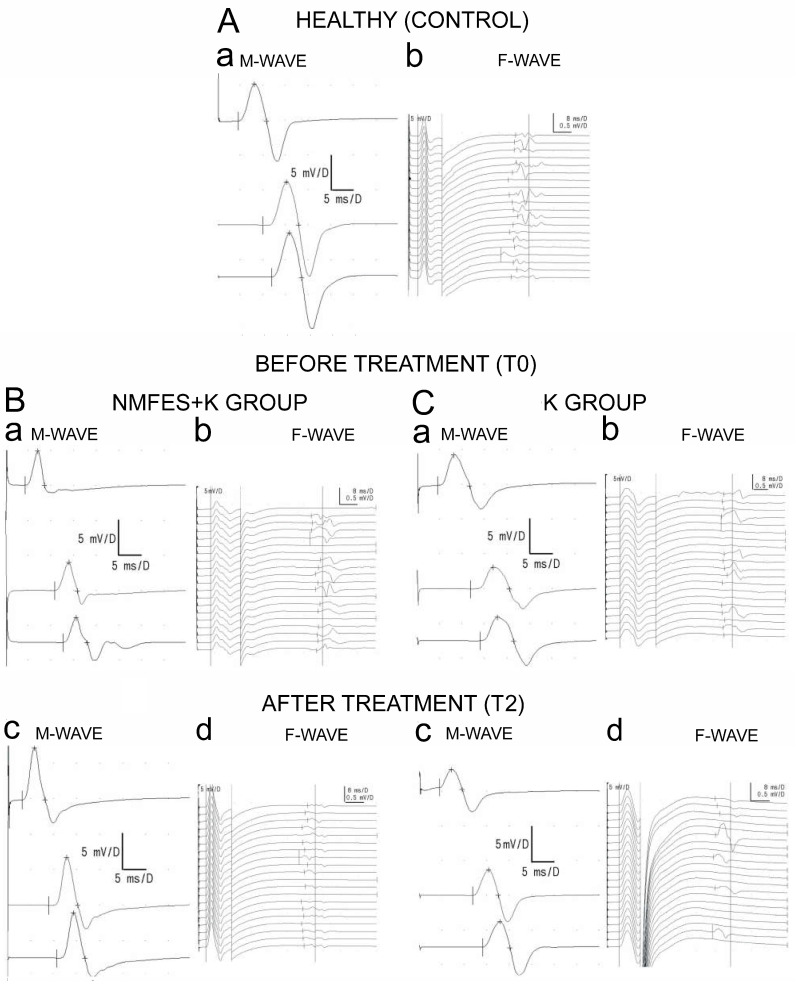
The examples of ENG recordings following stimulation of the peroneal nerves ((**a**), M-waves; (**b**), F-waves) performed in (**A**) a healthy volunteer, (**B**) one of the patients from the NMFES+K group, and (**C**) one of the patients from the K group. T0, before the treatment; and T2, after 60 days of treatment. Note the improvement of the M-wave amplitude at T2 observed in the patient from the NMFES+K group (**Bc**) and the development of the abnormality in the frequency of recorded F-waves in the patient from the K group also at T2 (**Cd**). Parameters of ENG recordings in **Bd** and **Cc** are similar to recorded in the healthy volunteers.

**Figure 6 ijerph-19-00713-f006:**
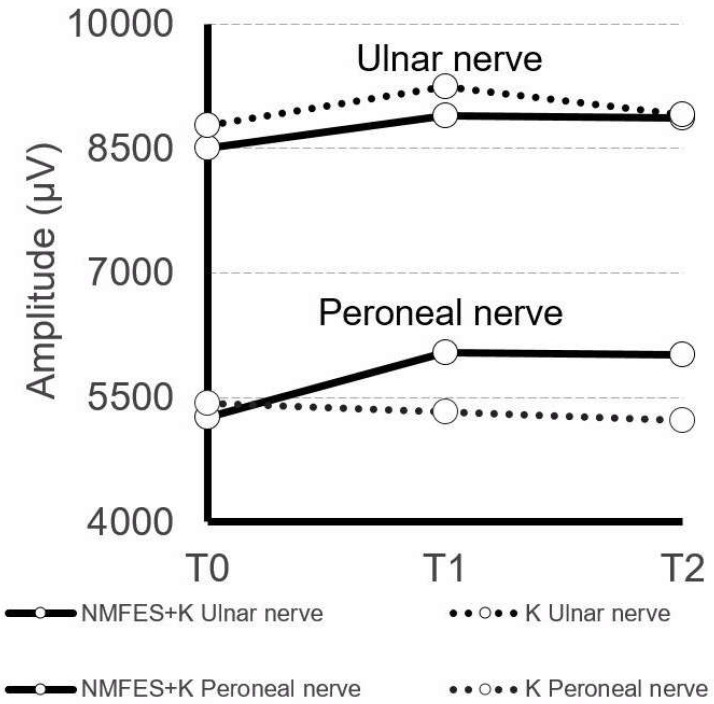
Summary of the neurophysiological results recorded in two groups of patients at three stages of observation.

**Table 1 ijerph-19-00713-t001:** Characteristics of the subjects.

Study Group Variable	Healthy Volunteers (Control) *N* = 60 41♀, 19♂		Patients NMFES+K Group *N* = 60 44♀, 16♂		Patients K Group *N* = 60 45♀, 15♂	
	Mean ± SD	Min–Max	Mean ± SD	Min–Max	Mean ± SD	Min–Max
Age	48.6 ± 4.3	30–52	62 ± 6.1	47–70	65 ± 5.2	56–70
Height (cm)	166.0 ± 4.8	161–180	16 3± 10.3	148–178	167 ± 7.2	157–180
Weight (kg)	75.3 ± 9.5	52–81	72 ± 11.1	55–95	74 ± 11.4	52–98
Observation time T0–T2 (days)	NA	NA	62 ± 6	50–72	63 ± 6	50–72

♀-female ♂-male.

**Table 2 ijerph-19-00713-t002:** Characteristics of the subjects and the summary of applied electrotherapy parameters.

Study Group Variable	Healthy Volunteers (Control) *N* = 60 41♀, 19♂		Patients NMFES+K Group *N* = 60 44♀, 16♂		Patients K Group *N* = 60 45♀, 15♂	
	Mean ± SD	Min-Max	Mean ± SD	Min-Max	Mean ± SD	Min-Max
Expected stimulation (hours)	NA	NA	19.2 ± 2.1	15–23	NA	NA
Detected stimulation (hours)	NA	NA	18.4 ± 4.3	16–24	NA	NA
Train stimulation frequency (Hz)	NA	NA	48.6 ± 6.1	35–70	NA	NA
Single stimulus duration (ms)	NA	NA	14.1 ± 15.2	12.5–17.5	NA	NA
Train duration (sec)	NA	NA	4.1 ± 1.7	3–6	NA	NA
Interval between trains (sec)	NA	NA	4.3 ± 1.2	2–5	NA	NA
Session duration (mins)	NA	NA	19.1 ± 2.2	15–20	NA	NA
Applied stimulus strength (mA) Upper extremity muscles -flexors -extensors	NA	NA	25.9 ± 3.1 26.2 ± 3.0	27–33 21–35	NA	NA
Applied stimulus strength (mA) Lower extremity muscles -flexors -extensors	NA	NA	25.4 ± 3.1 28.2 ± 3.3	21–37 23–32	NA	NA

NA––not applicable. ♀-female ♂-male.

**Table 3 ijerph-19-00713-t003:** Comparison of results from neurophysiological studies recorded in two groups of patients and healthy subjects. Results refer to measurements performed in patients on the paretic side identified in preliminary clinical examinations.

Test or Parameter	Healthy Volunteers *N* = 60	T0 Acute Phase (up to 7 Days after the Incident)		T1 Subacute phase (after 2–3 Weeks of Treatment)		T2 (after 2 Months of Rehabilitation Centre Treatment)		*p* Patients T0 vs. T2 before- -after	*p* Healthy vs. Patients T0 before	*p* Healthy vs. Patients T2 after
		Group NMFES+K Patients *N* = 60	Group K Patients *N* = 60	Group NMFES+K Patients *N* = 60	Group K Patients *N* = 60	Group NMFES+K Patients *N* = 60	Group K Patients *N* = 60			
ENG (M-waves parameters)										
Ulnar nerve Amplitude (µV)	9533 ± 2009	8580 ± 2227	8783 ± 2127	8809 ± 2122	9240 ± 2324	8872 ± 1175	8905 ± 2126	NMFES+K NS K NS	NMFES+K *p* = 0.05 K NS	NMFES+K NS K NS
Latency (ms)	3.5 ± 0.4	3.1 ± 0.8	3.0 ± 0.9	3.1 ± 0.7	3.2 ± 0.9	3.2 ± 0.8	3.2 ± 0.9	NMFES+K NS K NS	NMFES+K NS K NS	NMFES+K NS K NS
Peroneal nerve Amplitude (µV)	8677 ± 1122	5268 ± 1211	5432 ± 1125	6035 ± 1225	5321 ± 1028	6009 ± 928	5224 ± 1005	NMFES+K *p* = 0.04 K NS	NMFES+K *p* = 0.04 K *p* = 0.04	NMFES+K *p* = 0.04 K *p* = 0.04
Latency (ms)	4.5 ± 0.8	4.6 ± 0.9	4.7 ± 0.7	4.6 ± 0.7	4.5 ± 0.9	4.6 ± 0.9	4.6 ± 0.7	NMFES+K NS K NS	NMFES+K NS K NS	NMFES+K NS K NS
ENG (F-waves frequencies)										
Ulnar nerve Frequency	18 ± 1	17 ± 3	18 ± 2	18 ± 2	18 ± 2	19 ± 1	18 ± 2	NMFES+K NS K NS	NMFES+K NS K NS	NMFES+K NS K NS
Peroneal nerve Frequency	19 ± 1	17 ± 4	16 ± 3	16 ± 4	15 ± 5	16 ± 4	15 ± 5	NMFES+K NS K *p* = 0.05	NMFES+K *p* = 0.05 K *p* = 0.05	NMFES+K *p* = 0.05 K *p* = 0.04

## Data Availability

All the data generated or analyzed during this study are included in the published article.

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
