# Peer review of "Electroneurographic Evaluation of Neural Impulse Transmission in Patients after Ischemic Stroke Following Functional Electrical Stimulation of Antagonistic Muscles at Wrist and Ankle in Two-Month Follow-Up"

_ijerph, 2022, doi:10.3390/ijerph19020713_

Round 1

Reviewer 1 Report

Peer-review of the manuscript: ”Electroneurographical evaluation of neural impulses transmission in patients after ischemic stroke following functional electrical stimulation of antagonistic muscles at wrist and ankle in long term follow-up”

  1. General remarks:

The authors achieved a laborious clinical study that aimed to objectively compare, through electroneurographical evaluation of neural impulses transmission, the effectiveness of neuromuscular functional electrical stimulation (”NMFES”) applications on antagonistic muscle groups at the wrist and the ankle and kinesiotherapy based mainly on proprioceptive neuromuscular facilitation (PNF) … in post-stroke patients during ”long-term observation” – n. n.: ~ two months can be considered ”long-term” (?); this point of view must be either argued or to be avoided the mention ”long-term”.

  1. Specific remarks and suggestions:

- regarding the post stroke chronic phases: one of the bibliographic resources quoted just by the authors, i. e. Kraft et al. [7], refers to ”chronic (more than six months’ duration) stroke patients” !; however, 60 days/ two months (T2) is still in the subacute phase, not chronic ! (Dobkin BH, Carmichael ST. The Specific Requirements of Neural Repair Trials for Stroke. Neurorehabil Neural Repair. 2016 Jun;30(5):470-8. doi: 10.1177/1545968315604400. Epub 2015 Sep 10. PMID: 26359342; PMCID: PMC4786476.)

- ”plexopathies … and neuropathies” – plexopathies are not neuropathies ?

- based on which criteria – including from an ethical perspective: all the patients who met the including criteria, would presumably need/ benefit of the maximal type of interventions, i. e. ”NMFES + K” – some of the enrolled patients receive only K ?

- T0 was a pre-treatment moment, but precisely when – how many days after the (supra) acute stroke event – have started the treatment procedures: ”NMFES + K”, and respectively ”K” (?); some of them started as soon as 7 days after the (supra) acute stroke event, including with a 3 hours/ therapeutic-rehabilitative program ? Probably/ hopefully the enrolled patients were enough stable generally biological and neurological, so that the few deaths among them (”three patients of the K group and 3 patients of the NMFES+K group could not continue due to COVID-19 hospitalization or death”) had no connection with the duration of the (3 hours) therapeutic-rehabilitative program

- ”All the parameters were set up and supervised by a physiotherapist. The stimulating device was blocked to prevent unplanned changes applied by the patients up to T1, when the algorithm could be verified and modified by the physician.” So, the electro-stimulation (NMFES) parameters were ”set up and supervised”, ”verified and modified by” a Physical and Rehabilitation (PRM) physician or by just a physiotherapist (or the authors call the PRM physicians physiotherapists !?!)

- ”Electrostimulation was applied in an alternative mode.” This assertion needs to be detailed

- the applied electrical stimuli consisted of sinusoidal (i. e. with alternating positive and negative waves, resulting thus in cycles) or only of positive half waves/ (redressed) impulses ? – because, rigorously, only the alternating/ sinusoidal waves (having thus cycles) are measured in Hertz (Hz)

- ”orthopedic equipment (a walker, a quadriplegic)” ?

- ”the current study did not reveal changes in the upper extremity nerves on the paretic side. These differences might be explained by the dissimilar functional statuses of post-stroke patients, and the different structural symptoms of ischemic changes at the supraspinal level before the trials were undertaken.”  This explanation is not clear enough; it needs to be detailed, in order to be more precise/ convincing, and this goes for the following other one, regarding different results than of other studies: ”The convincing conclusion of Sahin et al. [18] and Guo et al. [19] about the effectiveness of NMFES together with PNF stretching on the wrist extensor muscle activity in poststroke patients may be explained by the facilitation of the two sources of afferent stimulations on the reflex motor recovery at the spinal or even the supraspinal levels. This explanation holds true, taking into account that the activity of the antagonistic muscle acting at the wrist and the ankle is regulated by the spinal Ia inhibitory interneurone reflex on a biofeedback way. The latter is, however, under the strong excitatory and inhibitory control of the impulses taking the origin from the corticospinal tract fibers and the propriospinal pathways [20,21].”

- ”therapist was able to set the parameters, save and secure the settings to prevent unplanned changes made by the participants. This had a significant impact on the patient's behavior, who, when under control, followed the stimulation regime more carefully.”  Aside the fact that throughout this manuscript – as commented above, too – there is an unclear, variable (and this has to be corrected for both reasons: terminological rigor and respect for the PRM Medical Specialty !) of the related professional naming (physiotherapist, or physician, or therapist !?!), the authors assert that yet, the patients could modify, by themselves, the ”stimulus strength” of their NMFES therapy (”The stimulus strength was the only parameter that could be changed by the participants. They were instructed to increase the stimulus strength during the single stimulation session to observe the visible contraction of the stimulated muscles and reach it without intrusive pain”); or, the electrical stimulation parameter: amplitude/ intensity (respectively its directly connected one: voltage) is a very important one, not negligible, which if modified it cannot be considered that this does not modify the ensemble of the ”stimulation regime”. This could also be an explanation of the partially different results obtained by the authors, compared to other studies with resembling focus. On the other hand – as the pain perception is a complex sense, for which different persons have, naturally, different thresholds – this could be a limitative factor within this study

- this last remark/ question refers also to the obvious difference (?) between the mean age of the control group and of the two groups of enrolled post-stroke patients

- Student's t -test is to compare mean values, and Mann–Whitney test is to compare median values, yet the latter type of values do not appear within the results ?

- Levene's test

Reviewer 2 Report

I really enjoyed reading your manuscript. I think that the field of this study will be helpful in finding an appropriate rehabilitation method for ischemic stroke patients in the future. I wrote down some comments and I hope they will help you in revising your manuscript.

Subjects and study design section
- Please indicate the gender of the study subject in table 1.

- Please describe in detail how the patient was assigned to the NMFES+K patient group and the K patient group(randomly assigned, etc.).

discussion section
- Please describe the limitations of the study.

Round 2

Reviewer 1 Report

I do appreciate the authors have seriously considered my observations, comments and suggestions - and made the related necessary corrections and improvements, as they have reported in their „Author response to report 1” - so I consider this new, corrected/ improved version of the manuscript is acceptable for publication, with just a single/ only correction I reckon it remained necessary; i. e., the authors corrected their initial formulation as follows: ”All the parameters were set up and supervised by a team of the physiotherapist and the physical medicine and rehabilitation physician.” In my opinion this is still not satisfactory because with all due respect, a physician - specifically in this case of the PRM specialty - is more academically qualified than a physiotherapist, and therefore he/she is legitimate/ entitled to lead such a team. Hence, I consider appropriate the authors would re-formulate this sentence like this: ”All the parameters were set up and supervised by a team of the physical and rehabilitation medicine physician and the physiotherapist.” 

Author Response

Dear Reviewer 1,

We greatly appreciate your positive review regarding our manuscript.

Please find the response below according to your suggestion.

A change has been highlighted with light blue in the text.

Remark 1. …”I do appreciate the authors have seriously considered my observations, comments and suggestions - and made the related necessary corrections and improvements, as they have reported in their „Author response to report 1” - so I consider this new, corrected/ improved version of the manuscript is acceptable for publication, with just a single/ only correction I reckon it remained necessary; i. e., the authors corrected their initial formulation as follows: ”All the parameters were set up and supervised by a team of the physiotherapist and the physical medicine and rehabilitation physician.” In my opinion this is still not satisfactory because with all due respect, a physician - specifically in this case of the PRM specialty - is more academically qualified than a physiotherapist, and therefore he/she is legitimate/ entitled to lead such a team. Hence, I consider appropriate the authors would re-formulate this sentence like this: ”All the parameters were set up and supervised by a team of the physical and rehabilitation medicine physician and the physiotherapist.””. “…

Response 1. Thank for this suggestion. We respect your expertise comment, therefore we have corrected the sentences in the lines 103,104; 192, 193; 324, 325; 360, 361   as follows:

All the parameters were set up and supervised by a team of the physical and rehabilitation medicine physician and the physiotherapist.

We hope that our explanation has answered to your remark good enough…

Sincerely yours,
Authors